# Robust Graph Structure Learning via Multiple Statistical Tests

**Yaohua Wang** [*]
Alibaba Group
xiachen.wyh@alibaba-inc.com

**Fangyi Zhang** [* †]
Queensland University of Technology
Centre for Robotics (QCR)
fangyi.zhang@qut.edu.au

**Ming Lin** [†]
Amazon
minglamz@amazon.com

**Senzhang Wang** [‡]
Central South University
szwang@csu.edu.cn

**Xiuyu Sun** [‡]
Alibaba Group
xiuyu.sxy@alibaba-inc.com

**Rong Jin** [†]
Twitter
rongjinemail@gmail.com

## Abstract

Graph structure learning aims to learn connectivity in a graph from data. It is particularly important for many computer vision related tasks since no explicit graph structure is available for images for most cases. A natural way to construct a graph among images is to treat each image as a node and assign pairwise image similarities as weights to corresponding edges. It is well known that pairwise similarities between images are sensitive to the noise in feature representations, leading to unreliable graph structures. We address this problem from the viewpoint of statistical tests. By viewing the feature vector of each node as an independent sample, the decision of whether creating an edge between two nodes based on their similarity in feature representation can be thought as a *single* statistical test. To improve the robustness in the decision of creating an edge, multiple samples are drawn and integrated by *multiple* statistical tests to generate a more reliable similarity measure, consequentially more reliable graph structure. The corresponding elegant matrix form named $\mathcal{B}$-**Attention** is designed for efficiency. The effectiveness of multiple tests for graph structure learning is verified both theoretically and empirically on multiple clustering and ReID benchmark datasets. Source codes are available at https://github.com/Thomas-wyh/B-Attention.

## 1 Introduction

Graph structure learning plays an important role in Graph Convolutional Networks (GCNs). It seeks to discover underlying graph structures for better graph representation learning when graph structures are unavailable, noisy or corrupted. These issues are typical in the field of vision tasks [58, 63, 62, 20, 42, 50, 57, 13, 39, 67, 47]. As an example, there is no explicit graph structure between photos in your phone albums, so the graph-based approaches [20, 42, 57] in photo management softwares seek to learn graph structure. A popular research direction on it is to consider graph structure

---

[*] Equal contribution.

[†] Work done while at Alibaba.

[‡] Corresponding author.

36th Conference on Neural Information Processing Systems (NeurIPS 2022).

learning as similarity measure learning upon the node embedding space [60]. They learn weighted adjacency matrices [20, 42, 9, 10, 43, 64, 28, 8, 25, 35] as graphs and feed them to GCNs to learn representations.

However, the main problem of the above weighted-based methods is that weights are not always accurate and reliable: two nodes with smaller associations may have a higher weight score than those with larger associations. The root of this problem is closely related to the **noise** in the high dimensional feature vectors extracted by the deep learning model, as similarities between nodes are usually decided based on their feature vectors. The direct consequence of using an inaccurate graph structure is so called feature pollution [57], leading to a performance degradation in downstream tasks. In our empirical studies, we found that close to $1/3$ of total weights are assigned to noisy edges when unnormalized cosine similarities between feature vectors are used to calculate weights.

Many methods have been developed to improve similarity measure between nodes in graph, including adding learnable parameters [9, 28, 8], introducing nonlinearity [10, 43, 64], Gaussian kernel functions [25, 35] and self-attention [53, 12, 20, 42]. Despite the progress, they are limited to modifying similarity function, without explicitly addressing the noise in feature vectors, a more fundamental issue for graph structure learning. One way to capture the noise in feature vectors is to view the feature vector of each node as an independent sample from some unknown distribution associated with each node. As a result, the similarity score between two nodes can be thought as the expectation of a test to decide if two nodes share the same distribution. Since, only one vector is sampled for each node in the above methods, the statistical test is essentially based on a single sample from both nodes, making the calculated weights inaccurate and unreliable.

An possible solution towards addressing the noise in feature vectors is to run multiple tests, a common approach to reduce variance in sampling and decision making [4, 11, 26]. Multiple samples are drawn for each node, and each sample can be used to run an independent test to decide if two nodes share the same distribution. Based on the results from **multiple tests**, we develop a novel and robust similarity measure that significantly improves the robustness and reliability of calculated weights. This paper will show *how* to accomplish multiple tests in the absence of graph structures and theoretically prove *why* the proposed approach can produce a better similarity measure under specified conditions. This similarity measure is also crafted into an efficient and elegant form of matrix, named $\mathcal{B}$-**Attention**. $\mathcal{B}$-**Attention** utilizes the fourth order statistics of the features, which is a significant difference with popular attentions. Benefiting from a better similarity measure produced by multiple tests, graph structures can be significantly improved, resulting in better graph representations and therefore boosting the performance of downstream tasks. We empirically verify the effectivness of the proposed method over the task of clustering and ReID.

In summary, this paper makes the following three major contributions: (1) To the best of our knowledge, this is the first paper to address inaccurate edge weights problem by statistical tests on the task of graph structure learning over images. (2) A novel and robust similarity measure is proposed based on multiple tests, proven theoretically to be superior and designed in an elegant matrix form to improve graph structure quality. (3) Extensive experiments further verify the robustness of the graph structure learning method against noise. State-of-the-art (SOTA) performances are achieved on multiple clustering and ReID benchmarks with the learned graph structures.

## 2 Methodology

To handle the noise in feature vectors, we model the feature vector of each node as a sample drawn from the underlying distribution associated with that node, and the similarity score between two nodes as the expectation of a statistical test. The **test** is to compare their similarity to a given threshold to decide if two nodes share the same distribution and that decision can be captured by a binary Bernoulli random variable. The comparison in statistical test may lead to error edges, i.e., **noisy edges** (an edge connects two nodes of different categories) or **missing edges** (two nodes of the same category should be connected but are not). The superiority of the similarity measure can be revealed by comparing the probability of creating an error edge. The proposed similarity measure will be introduced in Section 2.2, and its further design in matrix form will be presented in Section 2.3.

### 2.1 Preliminary

**Chernoff Bounds.** Sums of random variables have been studied for a long time [4, 11, 26]. Chernoff Bounds [11] provide sharply decreasing bounds that are in the exponential form on tail distributions

of sums of independent random variables. It is a widely used fundamental tool in mathematics and computer science [32, 41]. An often used and looser form of Chernoff Bounds is as follows.

**Lemma 1** (Chernoff Bounds [11]). *Suppose $X_i$ is a binary random variable taking value in set $\{0, 1\}$ with $\mathbb{P}(X_i = 1) = p_i$ . Define $X = \sum_1^n X_i$ where $\{X_1, X_2, \cdots, X_n\}$ are independently sampled. Then for any $0 < \epsilon < 1$,*

$$\mathbb{P}(X \geq (1 + \epsilon)\mu) \leq e^{-\frac{\epsilon^2}{3}\mu},$$

$$\mathbb{P}(X \leq (1 - \epsilon)\mu) \leq e^{-\frac{\epsilon^2}{3}\mu} .$$

It shows that with a high probability $1 - \delta$, we have $X \leq (1 + \epsilon)\mu$, or $X \geq (1 - \epsilon)\mu$, where $\delta = e^{-\frac{\epsilon^2}{3}}$.

**GCNs.** A GCN network normally consists of multiple GCN layers. Given an input graph $\mathcal{G}(\mathbf{X}, \mathbf{A})$ with $L$ vertices, the output of a normal GCN layer [34] is:

$$\mathbf{X}^{'} = \sigma(\tilde{\mathbf{D}}^{-\frac{1}{2}} \tilde{\mathbf{A}} \tilde{\mathbf{D}}^{-\frac{1}{2}} \mathbf{X} \mathbf{W}_l), \quad \text{where } \tilde{\mathbf{A}} = \mathbf{A} + \mathbf{I}, \ \tilde{\mathbf{D}}_{ii} = \sum_{j=1}^{L} \tilde{\mathbf{A}}_{i,j}, \tag{1}$$

$\mathbf{X} \in \mathbb{R}^{L \times M}$ is input vertex features with $M$ dimensions, $\mathbf{A} \in \mathbb{R}^{L \times L}$ is their adjacency matrix; $\mathbf{I} \in \mathbb{R}^{L \times L}$ is an identity matrix, $\tilde{\mathbf{D}} \in \mathbb{R}^{L \times L}$ is the diagonal degree matrix of $\tilde{\mathbf{A}}$; $\mathbf{W}_l \in \mathbb{R}^{M \times M'}$ is a trainable matrix, and $\sigma(\cdot)$ is an activation function such as ReLU [19]; $\mathbf{X}^{'} \in \mathbb{R}^{L \times M'}$ is the output vertex features with $M'$ dimensions, $M' = M$ in this work.

## 2.2 Multiple tests on similarity measure

Let $\mathcal{V} = \{v_1, v_2, ..., v_N\}$ be the collection of $N$ nodes. Each node is assumed to be in one of two categories: $+1$ or $-1$ for simplicity. For each node $v_i$, the subset of candidate nodes to connect it is denoted by $\mathcal{V}_i$. This subset can be decided by a simple criterion such as by $k$ nearest neighbours. When calculating the proposed similarity between $v_i$ and $v_j \in \mathcal{V}_i$, the common candidate nodes between $v_i$ and $v_j$, i.e., $\mathcal{V}_{i,j} = \mathcal{V}_i \cap \mathcal{V}_j$ are first identified. The nodes in the $\mathcal{V}_{i,j}$ are treated as the **multiple samples** drawn for the comparison of $v_i$ and $v_j$. Then, each $v_k \in \mathcal{V}_{i,j}$ performs single test twice, one for $v_k$ and $v_i$ and one for $v_k$ and $v_j$. The **single test** here is a comparison using the pairwise similarity based on the features of $v_k$ and $v_i$, or $v_k$ and $v_j$. The pairwise similarity can be cosine similarity, Gaussian kernel similarity or any other form, and is denoted by **Sim-S**. The proposed similarity score between $v_i$ and $v_j$ increases by one if the two tests yield $C(v_i) = C(v_k)$ and $C(v_k) = C(v_j)$, and zero otherwise, where $C(v)$ is the category of node $v$. It contains multiple single tests, hence the name **multiple tests**, corresponding similarity measure denoted by **Sim-M**.

For the convenience of analysis, a few assumptions are further introduced. First, the size of $\mathcal{V}_{i,j}$ is $m$, and among the nodes in $\mathcal{V}_{i,j}$, $\alpha m$ nodes share the same category as $v_i$ and $(1 - \alpha) m$ nodes belong to the opposite category, with $\alpha > \frac{1}{2}$. For any node $v_j \in \mathcal{V}_i$, its probability of being connected with $v_i$, according to the statistical test, is $q \in [0, 1]$ if $C(v_i) \neq C(v_j)$, and the probability increases to $1 \geq p > q$ if $C(v_i) = C(v_j)$. This section will show that, if $m$ is sufficiently large (i.e., enough number of tests can be run), there will exist an appropriate threshold for the Sim-M such that the number of error edges can be reduced significantly compared to Sim-S.

**Proposition 1.** *For the single test method, in expectation, the number of noisy edges $\mathbb{E}\{N_{\text{noisy}}\} = (1 - \alpha) qm$ and the number of missing edges $\mathbb{E}\{N_{\text{miss}}\} = \alpha (1 - p) m$.*

The theorem below shows that both $\mathbb{E}\{N_{\text{noisy}}\}$ and $\mathbb{E}\{N_{\text{miss}}\}$ can be reduced by a significant portion when $m$ is large enough.

**Theorem 1.** *Suppose $\gamma > 1$ and*

$$m > \frac{3((p + q)^2 + (2\alpha - 1)(p^2 - q^2))^2}{pq((p - q)^2 + (2\alpha - 1)(p^2 - q^2))^2} \log(\frac{\gamma}{\min((1 - \alpha)q, \alpha(1 - p))}), \tag{2}$$

*We have $\mathbb{E}\{N_{\text{noisy}}\} = (1 - \alpha) qm/\gamma$ and $\mathbb{E}\{N_{\text{miss}}\} = \alpha (1 - p) m/\gamma$.*

*Proof.* Consider node $v_i$ and one of its candidate node $v_j \in \mathcal{V}_i$. $S_{i,j}^k$ is denoted as the similarity score between $v_i$ and $v_j$ based on the statistical test with respect to $v_k$. As shown in Figure 1, $v_k$ may

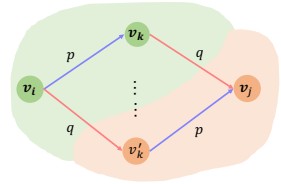

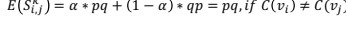

$E(S_{i,j}^k) = \alpha * pq + (1-\alpha) * qp = pq, if\ C(v_i) \neq C(v_j)$

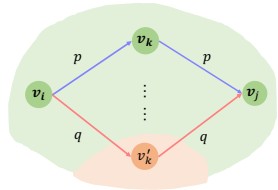

$E(S_{i,j}^k) = \alpha * p^2 + (1-\alpha) * q^2, if\ C(v_i) = C(v_j)$

Figure 1: Illustration of $\mathbb{E}[S_{i,j}^k]$ in Sim-M on two nodes from different categories and the same category. Nodes of different colors are from different categories.

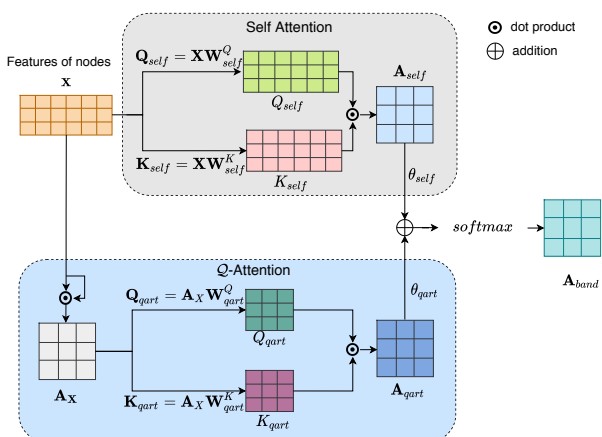

Figure 2: Illustration of the $\mathcal{B}$-**Attention** mechanism. The self-attention part is the same as that in Transformers. The $\mathcal{Q}$-**Attention** part generates $\mathbf{A}_X$ and then pay attention to it to generate the $\mathbf{A}_{qart}$. The two output attention maps are fused as the final output $\mathbf{A}_{band}$.

be of the same category as $v_i$ or not (denoted by $v'_k$ for a better show). Evidently, $\mathbb{E}[S_{i,j}^k] = pq$ if $C(v_i) \neq C(v_j)$, and $\mathbb{E}[S_{i,j}^k] = \alpha p^2 + (1-\alpha)q^2$ if $C(v_i) = C(v_j)$ .

The Sim-M between $v_i$ and $v_j$ is defined as:

$$S_{i,j} = \frac{1}{m} \sum_{k \in \mathcal{V}_{i,j}} S_{i,j}^k. \tag{3}$$

Using the Chernoff Bounds [11], we have with a probability $1 - \delta$,

$$\begin{cases} S_{i,j} = \frac{1}{m} \sum_{k \in V_{i,j}} S_k \leq (1 + \sqrt{\frac{3log(1/\delta)}{pqm}})pq, & C(v_i) \neq C(v_j), \\ S_{i,j} = \frac{1}{m} \sum_{k \in V_{i,j}} S_k \geq (1 - \sqrt{\frac{3log(1/\delta)}{(\alpha p^2 + (1-\alpha)q^2)m}})(\alpha p^2 + (1-\alpha)q^2), & C(v_i) = C(v_j), \end{cases} \tag{4}$$

By assuming that $m$ is large enough, i.e.,

$$\sqrt{\frac{3 \log(1/\delta)}{pqm}} < \frac{(p-q)^2 + (2\alpha - 1)(p^2 - q^2)}{(p+q)^2 + (2\alpha - 1)(p^2 - q^2)}, \tag{5}$$

by choosing the similarity threshold

$$S_t = (1 + \sqrt{\frac{3log(1/\delta)}{pqm}})\frac{pq}{2} + (1 - \sqrt{\frac{3log(1/\delta)}{(\alpha p^2 + (1-\alpha)q^2)m}})\frac{\alpha p^2 + (1-\alpha)q^2}{2}, \tag{6}$$

only with a probability $\delta$, we will miss a correct edge, and with a probability $\delta$ we will introduce a noisy edge. We complete the proof by substituting $\delta$ in Equation 5 with

$$\delta \triangleq \frac{1}{\gamma} \min((1-\alpha)q, \alpha(1-p)). \tag{7}$$

We can guarantee that the Sim-M between two nodes from the same category is strictly larger than that for two nodes from different categories, leading to the distillation of all noisy edges. $\qquad \square$

## 2.3 Multiple tests on GCNs

With a better similarity measure prototype above, we further craft it into an elegant and efficient matrix form for adaptation of graph structure learning in the real-world. The analysis in **Theorem** 1 can be

easily extended to real similarities, instead of binary simiarities, by using the generalized version of Chernoff inequality detailed in Section E of Appendix. Given the L2 normalized original node features $\mathbf{X}$, the idea of Sim-S can be implemented by inner product of features, i.e. $\mathbf{A}_X = \mathbf{X}\mathbf{X}^T$. The twice-run tests on each node can be achieved by $\mathbf{A}_X$ multiplying itself. Inspired by the design of popular Transformers [53], some learnable parameters are also introduced. It contains the **fourth-order statistics of features**, which is a significant difference, since self-attention only has second-order. If self-attention is regarded as **Duet of Attention**, our method is analogous to **Quertet of Attention** ($\mathcal{Q}$-**Attention** for short). The vectorized representation of Equation 3, $\mathcal{Q}$-Attention $\mathbf{A}_{qart} \in \mathbb{R}^{L \times L}$, is then implemented as follows:

$$\mathbf{A}_{qart} = \mathbf{Q}_{qart}\mathbf{K}_{qart}^T, \quad \text{where } \mathbf{Q}_{qart} = \mathbf{A}_X\mathbf{W}_{qart}^Q, \ \mathbf{K}_{qart} = \mathbf{A}_X\mathbf{W}_{qart}^K, \tag{8}$$

$\mathbf{W}_{qart}^Q \in \mathbb{R}^{L \times L'}$ and $\mathbf{W}_{qart}^K \in \mathbb{R}^{L \times L'}$ are learnable weights, $L' = L$ in this work. L2 normalization is also applied to $\mathbf{A}_X$.

Motivated by the further enhancements from combinations of intrinsic graph structures and implicit graph structures in literature [35, 8, 36], the self-attention can play an intrinsic role to fuse with the $\mathcal{Q}$-**Attention** in vision tasks, although no intrinsic graph structure exists between images at all. The **self-attention** part is the same as that in Transformers [53]. The attention map $\mathbf{A}_{self} \in \mathbb{R}^{L \times L}$ is:

$$\mathbf{A}_{self} = \frac{\mathbf{Q}_{self}\mathbf{K}_{self}^T}{\sqrt{M^d}}, \quad \text{where } \mathbf{Q}_{self} = \mathbf{X}\mathbf{W}_{self}^Q, \ \mathbf{K}_{self} = \mathbf{X}\mathbf{W}_{self}^K, \tag{9}$$

$\mathbf{W}_{self}^Q \in \mathbb{R}^{M \times M^d}$ and $\mathbf{W}_{self}^K \in \mathbb{R}^{M \times M^d}$ are the learnable weights, $M^d$ is the dimension of query and key features. The combined form blends the duet and quartet of attention, hence the name **Band of Attention** ($\mathcal{B}$-**Attention** for short). The overview of $\mathcal{B}$-**Attention** mechanism is shown in Figure 2.

The fusion of $\mathbf{A}_{self}$ and $\mathbf{A}_{qart}$ is defined as (where softmax($\cdot$) is a softmax function):

$$\mathbf{A}_{band} = \text{softmax}(\theta_{qart}\mathbf{A}_{qart} + \theta_{self}\mathbf{A}_{self}). \tag{10}$$

When combined with the neural network layer in Equation 1, the final form is:

$$\mathbf{X}' = \sigma(\mathbf{A}_{band}\mathbf{X}\mathbf{W}_l). \tag{11}$$

In real implementation, considering the graph size can be very large for large-scale datasets, sub-graph sampling strategy [57, 55] is adopted to make it more computationally efficient and scalable. In particular, $k_{seed}$ nodes are first selected as seeds from a dataset in each sampling step. Then the seed nodes together with their $k$ nearest neighbours ($k$NN) [14] form the nodes of a sub-graph. As a result, Equations 8 and 9 are applied to the sub-graph during training and inference. For a dataset of size $N_d$, it takes about $N_d/k_{seed}$ iterations to process the whole dataset on average. The pseudo code for self-attention and $\mathcal{Q}$-**Attention** is detailed in Section D of Appendix.

## 3 Experiments

Experiments are designed with three levels. Firstly, experiments are conducted to analyse the advantage of Sim-M over that of Sim-S and its robustness to noise. Secondly, armed with multiple tests on GCNs, i.e., $\mathcal{B}$-**Attention** is further investigated in comparison with the self-attention and a commonly used form of GCNs on robustness and superiority. Ablation experiments are then conducted to study how each part of $\mathcal{B}$-**Attention** contributes to and influences the performance. Finally, with the robust graph structure learned by $\mathcal{B}$-**Attention**, comparison experiments in downstream graph-based clustering and ReID tasks are conducted to evaluate the benefits of the proposed approach.

### 3.1 Experimental setups

**Datasets** Experiments are conducted on commonly used visual datasets[4] with three different types of objects, i.e., **MS-Celeb** [21] (human faces), **MSMT17** [59] (human bodies) and **VeRi-776** [37] (vehicles), to verify the generalization of the proposed method. In these datasets, each sample has a specific category, and most categories have rich samples. The tasks on these datasets are popular and representative in computer vision. It is worth noting that MS-Celeb is divided into 10 parts by identities (Part0-9): Part0 for training and Part1-9 for testing, to maintain comparability and consistency with previous works [63, 63, 20, 57, 42, 50]. **Details of the dataset** are in Section A.1.

---

[4]**DECLARATION:** Considering the ethical and privacy issues, all the datasets used in this paper are feature vectors which can not be restored to images.

**Metrics** The Area Under the Curve (**AUC**) is adopted to directly evaluate the discriminatory power of similarity measures between node pairs. Edge Noise Rate (**ENR**) illustrates the amount of noise in graphs quantitatively. For node $v_i$, $\mathbf{ENR} = \frac{N_i^{noise}}{N_i^{all}}$, where $N_i^{noise}$ is the number of noisy edges and $N_i^{all}$ is the amount of all edges for $v_i$. The average ENR is defined as the average of the ENRs of all nodes. The feature quality of GCNs output is investigated to prove the effectiveness of graph structure learning: The mean Average Precision (**mAP**) [68] is used from the perspective of identification and **ROC** curves are shown from the perspective of verification. The higher the quality of the features the better the graph structure learning. In downstream ReID tasks, $\mathbf{mAP^R}$ and Rank1 accuracy (**R1**) are used. Compared to mAP, $\mathbf{mAP^R}$ additionally takes into account the camera id information, i.e., samples from the same camera as the probe will not be counted in evaluation. **BCubed** ($\mathbf{F_B}$) [3, 1] and **Pairwise** ($\mathbf{F_P}$) [51] F-scores are used for evaluating clustering tasks.

## 3.2 Effectiveness of the Sim-M

To verify the robustness of Sim-M, the study is first conducted on MS-Celeb (Part1) by evaluating the discriminatory ability on different average ENRs comparing with Sim-S. The Sim-S uses the cosine similarity here for simplicity and without loss of generality. For the sake of sparsity and efficiency it brings, the graphs are built based on $k$NN via the approximate nearest neighbour algorithm. The average ENR of graphs is controlled by using different $k$ values when searching neighbours. AUC is adopted to evaluate the similarity score between node pairs which have $k$NN relations. Table 1 presents that as the $k$ increases, more and more noise is added (ENR gets larger). For Sim-M, however, more samples can be drawn and more tests can be run, so the advantage of Sim-M over Sim-S becomes increasingly obvious. This demonstrates the **robustness** of Sim-M to noise. This phenomenon also conforms

Table 1: Average ENRs of different $k$ values and the AUC of Sim-S ($\mathrm{AUC_S}$), Sim-M ($\mathrm{AUC_M}$) and their difference ($\mathrm{AUC_\delta}=\mathrm{AUC_M}$-$\mathrm{AUC_S}$) on MS-Celeb (part1). The values of AUC are scaled by a factor of 100 for better display.

| $k$ | ENR | $\mathrm{AUC_S}$ | $\mathrm{AUC_M}$ | $\mathrm{AUC_\delta}$ |
|---|---|---|---|---|
| 5 | 0.05 | 96.80 | 97.59 | 0.79 |
| 10 | 0.09 | 96.21 | 97.72 | 1.51 |
| 20 | 0.14 | 95.33 | 97.21 | 1.88 |
| 40 | 0.23 | 94.00 | 95.66 | 1.66 |
| 80 | 0.39 | 92.05 | 91.32 | -0.71 |
| 120 | 0.54 | 90.76 | 86.58 | -4.17 |
| 160 | 0.64 | 90.65 | 83.90 | -6.75 |

to the condition and conclusion in **Theorem** 1. However, as $k$ increases and ENR gets larger, too much noise can not be handled by multiple tests, so the performance of Sim-M begins to drop. This downward trend can be slowed down by the careful design of $\mathcal{B}$-**Attention**. The same experiments are conducted on the MSMT17 and VeRi-776, with similar findings in Section A.2.

## 3.3 Effectiveness of the $\mathcal{B}$-Attention

To demonstrate the robustness of $\mathcal{B}$-**Attention** mechanism, the study is first conducted on MS-Celeb (Part1) by evaluating the performance of $\mathcal{B}$-**Attention** with different average ENRs, comparing it to other baselines. Then, comparison experiments are also conducted on MSMT17, VeRi-776 and also larger scales (Part3-9) of MS-Celeb with the best ENR settings for each dataset. Comparison baselines include the original features of each dataset, a commonly used form of GCN [34, 57], Transformer, GCN with the self-attention (GCN with $\mathbf{A}_{self}$), Transformer with the $\mathcal{B}$-**Attention** mechanism (Trans with $\mathbf{A}_{band}$), and GCN with $\mathcal{B}$-**Attention** mechanism (GCN with $\mathbf{A}_{band}$). Transformer here is only used as an encoder for structure learning, no positional encoding is required.

**Network architectures and settings** The model is composed of multiple GCN layers with $\mathcal{B}$-**Attention**, and an extra fully connected layer with PReLU [23] activation. The output features have a dimension of 2048 for all datasets. The number of GCN layers is also tuned respectively for different datasets: MS-Celeb uses three GCN layers; MSMT17 and VeRi-776 use one GCN layer. Each node acts as the probe to search its $k$NN to construct the graph. The $k$ value is tuned respectively for different datasets: 120 for MS-Celeb, 30 for MSMT17, and 80 for VeRi-776. Each node in a dataset can be enhanced multiple times, as either a probe or a neighbour. The Hinge Loss [45, 57] is used to classify node pairs. The initial learning rate is 0.008 with the cosine annealing strategy.

**Performance on different average ENRs** Models are trained on Part0 and tested on Part1 of MS-Celeb. All baselines use three GCN/Transformer layers for MS-Celeb, except the GCN with

$\mathbf{A}_{self}$ which uses two GCN layers (observed to have better performance than three GCN layers). For a certain ENR, the same $k$ value is used in training and testing. Figure 3 shows how the output feature qualities (mAP) change with different average ENRs for all baselines. It shows that the GCN and Transformer with $\mathbf{A}_{band}$ work the best in most average ENRs in terms of both mAP values and their robustness to different ENRs. The GCN has the worst results, and the performances of the ones with $\mathbf{A}_{self}$ are in between. This validates that a commonly used form of GCN is sensitive to noisy edges in input graphs; the proposed $\mathcal{B}$-**Attention** can well alleviate the influence of noisy edges and show the strongest robustness when learning graph structure; the $\mathbf{A}_{self}$ also has some power of dealing with noisy edges, but it is worse than $\mathbf{A}_{band}$. When $k$ value is too small (e.g., 5, 10), the performances of $\mathbf{A}_{band}$ and $\mathbf{A}_{self}$ are similar. This indicates that the $\mathcal{Q}$-**Attention** part of $\mathcal{B}$-**Attention** can hardly bring a positive effect when there are too few context neighbours, i.e. the number of single tests gathered is insufficient. In addition, GCN and Transformer have very similar performance, either with $\mathbf{A}_{band}$ or $\mathbf{A}_{self}$. This indicates that base networks are not critical, and both attention mechanisms work for the two base networks. The performance degradation of $\mathcal{B}$-**Attention** with very large ENR (e.g., 0.64) indicates that the current form $\mathcal{B}$-**Attention** still has improvement room, although works much better than the self-attention. Some empirical case studies on self attention, $\mathcal{Q}$-**Attention** and $\mathcal{B}$-**Attention** are shown in Section F of Appendix to better understand the effect of multiple tests.

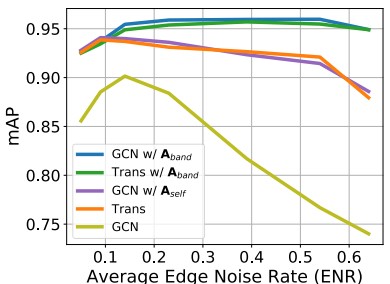

Figure 3: Feature quality (mAP) of various methods with different average ENRs on MS-Celeb Part1.

Table 2: Feature quality (mAP) of different methods on visual datasets of different types of objects.

| Dataset | MS-Celeb | MSMT17 | VeRi-776 |
|---|---|---|---|
| Original Feature | 80.07 | 74.46 | 81.91 |
| GCN | 90.15 | 83.51 | 85.27 |
| Transformer | 93.86 | 85.44 | 85.92 |
| GCN w/ $\mathbf{A}_{self}$ | 94.09 | 85.40 | 86.88 |
| Trans w/ $\mathbf{A}_{band}$ | 95.70 | 85.92 | 87.07 |
| GCN w/ $\mathbf{A}_{band}$ | **95.97** | **86.04** | **87.59** |

**Performance on different types of objects** To further validate the $\mathcal{B}$-**Attention** mechanism, comparisons are also made on datasets with other types of objects: MSMT17 and VeRi-776. In these two datasets, $k$ values and network depths are also **tuned** for all baselines, aiming for their best performances. For both datasets, one GCN layer is used for GCN, GCN with $\mathbf{A}_{band}$, and Transformer with $\mathbf{A}_{band}$; three GCN layers are used for GCN or Transformer with $\mathbf{A}_{self}$. In MSMT17, $k$=10 for GCN and "GCN w/ $\mathbf{A}_{self}$"; $k$=20 for Transformer; $k$=30 for the two $\mathcal{B}$-**Attention** methods. In VeRi-776, $k$=80 for "GCN w/ $\mathbf{A}_{band}$" and "Trans w/ $\mathbf{A}_{band}$", and $k$=20 for the rest of baselines. The best mAP results of all baselines are listed in Table 2, where the results for MS-Celeb are the peak values in Figure 3. From Table 2, the $\mathcal{B}$-**Attention** works the best in all three datasets; $\mathbf{A}_{self}$ works better than GCN; and all output features are much better than the original ones, although the original features of MSMT17 and VeRi-776 are current SOTA ones. The best performances are achieved by the GCN with $\mathbf{A}_{band}$ in all datasets. Their ROC curves are shown in Figure 4, where the same observations can be seen from verification perspective. As shown in Figure 4 and Table 2, the performance differences among those baselines with attention mechanisms are less significant in MSMT17 where $k$ values are relatively small and close across baselines. This observation is similar to that in Figure 3: the performance margin between $\mathbf{A}_{band}$ and $\mathbf{A}_{self}$ is less significant when average ENRs (i.e., $k$) are relatively small. This further shows that the benefit of $\mathcal{B}$-**Attention** is limited when with a small number of neighbours, as not many single tests to run and leverage. Better performance can not be achieved by enlarging $k$ values for MSMT17, as its average sample amount for each individual is only 31. Our analysis assumes that for any node $v$, there are more nodes in the $k$ nearest neighbors sharing the same category as $v$ than those sharing different categories (i.e. $\alpha > \frac{1}{2}$). By significantly enlarging $k$, we will end up with $\alpha < \frac{1}{2}$ in $k$ nearest neighbors, which fails the assumption of our algorithm. Too large $k$ values would result in too big ENR which can not be handled by the current form of $\mathcal{B}$-**Attention** as discussed on Figure 3.

Experiments are also conducted to compare these baselines on **larger scales of MS-Celeb** (i.e., Part3-9) in Section A.3 of Appendix. Results in Table 6 and Figure 5 show that all baselines

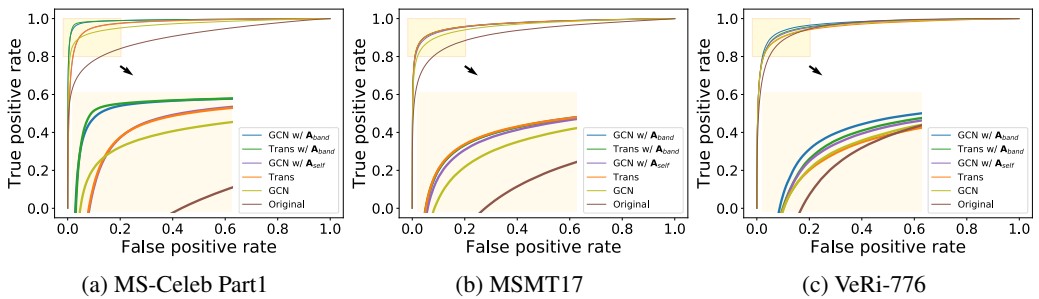

Figure 4: ROC curves of different methods on MS-Celeb Part1 (a), MSMT17 (b) and VeRi-776 (c).

Table 3: Clustering performance on MS-Celeb.

| Dataset | Part1 | | Part3 | | Part5 | | Part7 | | Part9 | |
|---|---|---|---|---|---|---|---|---|---|---|
| Metrics | $F_P$ | $F_B$ | $F_P$ | $F_B$ | $F_P$ | $F_B$ | $F_P$ | $F_B$ | $F_P$ | $F_B$ |
| K-Means | 79.21 | 81.23 | 73.04 | 75.20 | 69.83 | 72.34 | 67.90 | 70.57 | 66.47 | 69.42 |
| HAC | 70.63 | 70.46 | 54.40 | 69.53 | 11.08 | 68.62 | 1.40 | 67.69 | 0.37 | 66.96 |
| DBSCAN | 67.93 | 67.17 | 63.41 | 66.53 | 52.50 | 66.26 | 45.24 | 44.87 | 44.94 | 44.74 |
| Clusformer | 88.20 | 87.17 | 84.60 | 84.05 | 82.79 | 82.30 | 81.03 | 80.51 | 79.91 | 79.95 |
| STAR-FC | 91.97 | 90.21 | 88.28 | 86.26 | 86.17 | 84.13 | 84.70 | 82.63 | 83.46 | 81.47 |
| Ada-NETS | 92.79 | 91.40 | 89.33 | 87.98 | 87.50 | 86.03 | 85.40 | 84.48 | 83.99 | 83.28 |
| Original+G-cut | 69.63 | 73.62 | 63.31 | 66.56 | 60.61 | 62.91 | 57.13 | 58.72 | 54.42 | 58.54 |
| Ours+G-cut | **93.90** | **92.47** | **90.54** | **89.23** | **88.70** | **87.13** | **86.18** | **85.59** | **84.36** | **84.10** |
| Original+Infomap | 93.91 | 92.42 | 90.10 | 89.08 | 88.49 | 87.30 | 85.31 | 86.01 | 83.08 | 85.01 |
| Ours+Infomap | **94.94** | **93.67** | **91.74** | **90.81** | **89.50** | **89.15** | **87.04** | **87.81** | **85.40** | **86.76** |

have performance degradation when the scale enlarges while the degradation of that "w/ $\mathbf{A}_{band}$" is much smaller than baselines. These results for the cases with different ENRs, on different types of objects and larger scales of testsets, together validate the robustness and effectiveness of proposed $\mathcal{B}$-**Attention** in graph structure learning.

**Ablation experiments** on the design of the topological architecture and attention fusion method are also conducted in Section A.4 of Appendix.

## 3.4  Effectiveness of the downstream tasks

The improvement of graph structure learning can also be proven through the downstream tasks by exploiting the features it produces. The better the graph structure is learned, the better the performance on downstream tasks. Graphs have important applications in large-scale clustering and re-reranking in ReID. However, they suffer from inaccurate weight problems when learning and working with the graph structure. Armed the GCN with $\mathbf{A}_{band}$, many SOTA results are achieved.

**Performance on clustering tasks**  The clustering tasks are evaluated on MS-Celeb and MSMT17. Both of them have a large number of classes and are suitable as clustering benchmarks. The same features output by "GCN w/ $\mathbf{A}_{band}$" are used as that in Section A.3 for MS-Celeb and Section 3.3 for MSMT17. The same clustering method (denoted "G-cut") is adopted as that in [20, 57], where pairs of nodes are linked when their similarity scores are larger than a tuned threshold, and clustering is finally completed by transitively merging all links via a union-find algorithm [2]. The threshold is tuned for each dataset, aiming for the best balance between BCubed and Pairwise F-Scores. In addition, map equation based unsupervised clustering algorithm Infomap [46, 61] is also adopted here. Their BCubed and Pairwise F-scores are listed in Table 3 and Table 10, where the results of current SOTA GCN/Transformer methods (Clusformer [42], STAR-FC [50] and Ada-NETS [57]) and three conventional clustering approaches (K-Means [38], HAC [52] and DBSCAN [17]) are also listed. The results show that the features enhanced via our approach have the best clustering results in all datasets, either with G-cut or Infomap, both of which outperform the current SOTA methods Ada-NETS and STAR-FC with large margins. In comparison, the results of the ones with Infomap are better than those with G-cut.

**Performance on ReID tasks** The ReID tasks is evaluated on two ReID datasets: MSMT17 and VeRi-776. The enhanced features used are the same as those in Table 2. mAP$^\mathbf{R}$ and R1 are evaluated following the standard probe and gallery protocols of MSMT17 and VeRi-776. The results are shown in Table 4. As the output feature of learned graph can also be understood as a rerank process in ReID, comparisons are also made on commonly used rerank methods: QE [13], LBR [39], KR [67] and ECN [47]. Table 4 shows that the "GCN w/ $\mathbf{A}_{band}$" achieves the best or comparable performance, and all enhanced features are much better than the original SOTA ones.

Table 4: ReID performance on MSMT17 and VeRi-776.

| Datasets | MSMT17 | | VeRi-776 | |
|---|---|---|---|---|
| Metrics | mAP$^R$ | R1 | mAP$^R$ | R1 |
| Original Feature | 62.76 | 82.36 | 79.00 | 96.60 |
| GCN | 74.66 | 85.36 | 82.31 | 96.60 |
| Transformer | 77.60 | 86.37 | 82.62 | 96.54 |
| GCN w/ $\mathbf{A}_{self}$ | 77.34 | 85.49 | 84.29 | 96.30 |
| Trans w/ $\mathbf{A}_{band}$ | 77.49 | 86.54 | 83.98 | 96.72 |
| GCN w/ $\mathbf{A}_{band}$ | 78.28 | **86.64** | **84.74** | 96.78 |
| Original+QE | 71.10 | 84.37 | 83.09 | 95.47 |
| Original+LBR | 70.36 | 85.28 | 83.63 | **97.13** |
| Original+KR | 76.46 | 85.48 | 82.32 | 96.84 |
| Original+ECN | **78.79** | 86.16 | 84.25 | **97.13** |

In MSMT17, similar to the results in Section 3.3, the baselines with attention mechanism have very close performance, but work much better than the GCN. In VeRi-776, from the mAP$^\mathbf{R}$ perspective, $\mathcal{B}$-**Attention** works better than $\mathbf{A}_{self}$, and both of them are better than the GCN, but their R1 results are comparable. These results still support the superiority of the proposed approach, although R1 results of VeRi-776 are comparable, as mAP$^\mathbf{R}$ is a more comprehensive indication of feature quality for ReID. The experiments on the **combination** of "GCN w/ $\mathbf{A}_{band}$" with these commonly used rerank methods are also conducted in Section A.6, achieving more SOTA performances and proving the complementarity between them.

# 4 Related Work

**Graph Convolutional Networks.** GCNs [34, 22, 54] extends the operation of convolution from the grid-data (e.g. images, videos) to non-grid data which is more common in the real world. The classic GCN [34] is proposed based on the spectral theory and achieves promising results on citation network datasets: Citeseer, Cora and Pubmed [49]. GraphSAGE [22] further extends GCNs from transductive framework to inductive framework and obtains better results. GAT [54] introduces the attention mechanism in GCN, making it more expressive. Fast-GCN [6] interprets graph convolutions as integral transforms of embedding functions under probability measures, which not only is efficient for training but also generalizes well for inference. Some research works [48, 66, 27, 56, 7] also advance GCNs for relational data modeling. However, these GCNs are all investigated on relational datasets that have explicit graph structures. This work aims to extend the capacity of GCNs to work on datasets (e.g., image objects) which have no explicitly defined graph structures.

**Graph Structure Learning.** The literature [60] suggests that research on graph structure learning for GCNs can be roughly divided into discrete-based and weighted-based ones. Some discrete-based methods [16, 31, 40, 18] treat the graph structures as random distributions and sample binary adjacency matrices from them. Another way [57] constructs discrete graphs based on heuristic proxy objectiveness aiming for clean yet rich neighbours for each node. The binary matrices methods are difficult to optimize because of the gradient breakage, so such methods resort to variational inference (e.g., approaches in [16, 40]), reinforcement learning (e.g., approaches in [31]) or optimization in stages (e.g., approaches in [57]), which may lead to sub-optimal solutions. In addition, the parameter size of distributions [18] can be very large (e.g., $\mathcal{O}(Nk)$ where $N$ is the number of nodes and $k$ is the number of neighbours for each node) and therefore not suitable for large datasets. The weighted-based methods [9, 10, 43, 64, 28, 8, 25, 35, 20, 42], which are the focus of this paper, learn weighted adjacency matrices so that it can be optimized by SGD techniques [44, 33, 5]. The main difference between these works [9, 10, 43, 64, 28, 8, 25, 35] is in the similarity measure, which can be classified into attention-based [9, 10, 43, 64, 28, 8] and kernel-based [25, 35] roughly. To enhance the discriminatory power of the similarity measure, they have adopted various forms: adding learnable parameters [9, 28, 8], introducing nonlinearity [10, 43, 64], Gaussian kernel functions [25, 35] or self-attention [20, 42, 53, 12] etc. However, these methods are sensitive to the noise from feature representations, particular when each node is represented by a high dimensional vector, leading to the problem of inaccurate weights and vulnerable graph structure.

This work aims to propose a multiple samples based similarity measure to improve the robustness of graph structure learning. Another approach uses the low-rank, sparsity and feature smoothness properties of the real-world graph to optimize the graph structure [29], but these priors are suitable for community networks rather than graph of image objects. Some work [30] learns to augment the given graph with new edges by reinforcement learning to improve the performance of downstream tasks, but existing noisy edges problem in graphs cannot be handled.

## 5 Conclusion

The problem of inaccurate edge weights is identified in graph structure learning, analysed from the perspective of noise in feature vectors, and handled based on several tests. A novel and robust similarity measure Sim-M is proposed, proven to be superior and verified in experiments. Then it is designed in an elegant matrix form, i.e., $\mathcal{B}$-**Attention** mechanism, to improve the graph structure quality in GCNs. The superiority and robustness of $\mathcal{B}$-**Attention** are validated by comprehensive experiments and achieve many SOTA results on clustering and ReID benchmarks.

## Acknowledgments and Disclosure of Funding

We thank many colleagues at DAMO Academy of Alibaba, in particular, Yuqi Zhang, Weihua Chen and Hao Luo for useful discussions.

This research was partially funded by the National Science Foundation of China (No. 62172443) and Hunan Provincial Natural Science Foundation of China (No. 2022JJ30053).

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
