# A  Experiments

## A.1  Details of the dataset

MS-Celeb [21] is a dataset with 10M face images of nearly 100K individuals. A cleaned version [15] with about 5.8M face images of 86K identities is used in this paper. The protocol and features are the same as that in [63], where the dataset is divided into 10 parts by identities (Part0-9): Part0 for training and Part1-9 for testing. The feature dimension is 256. The data amounts of Part 0-9 (0, 1, 3, 5, 7 and 9) are respectively: 584K, 584K, 1.74M, 2.89M, 4.05M and 5.21M.

MSMT17 [59] is the current largest ReID dataset. It contains 32,621 images of 1,041 individuals for training and 93,820 images of 3,060 individuals for testing. These images are captured by 15 cameras under different time periods, weather and light conditions. This work uses the SOTA features from AGW [65] with a dimension of 2048.

VeRi-776 [37] is a commonly used vehicle ReID benchmark, which contains over 40K bounding boxes of 619 vehicles captured by 20 cameras in unconstrained traffic scenes. The dataset is divided into two parts: 37,715 images of 576 vehicles for training, and 13,257 images of 200 vehicles for testing. This work uses the features extracted by the SOTA ViT-B/16 Baseline in TransReID [24], with 768 dimensions.

## A.2  Effectiveness of the Sim-M

The experiments on MSMT17 and VeRi-776 also show that the advantage of Sim-M over Sim-S is increasingly obvious as $k$ increases and then drops for introduction of too much noise as in Table 5.

Table 5: AUC of Sim-S ($AUC_S$), Sim-M ($AUC_M$) and their difference ($AUC_\delta = AUC_M - AUC_S$) on MSMT17 and VeRi-776. The values of AUC are scaled by a factor of 100 for a better show.

| Datasets | MSMT17 | | | | VeRi-776 | | | |
|---|---|---|---|---|---|---|---|---|
| $k$ | ENR | $AUC_S$ | $AUC_M$ | $AUC_\delta$ | ENR | $AUC_S$ | $AUC_M$ | $AUC_\delta$ |
| 5 | 0.06 | 89.23 | 89.74 | 0.52 | 0.01 | 90.96 | 90.40 | -0.56 |
| 10 | 0.13 | 89.33 | 90.02 | 0.69 | 0.03 | 89.34 | 90.09 | 0.75 |
| 20 | 0.26 | 87.78 | 88.63 | 0.85 | 0.07 | 85.86 | 86.54 | 0.68 |
| 40 | 0.48 | 86.72 | 86.62 | -0.10 | 0.17 | 83.05 | 82.89 | -0.16 |

## A.3  Performance on larger scales of MS-Celeb

Experiments are also conducted to compare these baselines on larger scales of MS-Celeb (i.e., Part3-9). The same model for MS-Celeb Part1 is used as in Section 3.3, but tested on Part3-9. The results in Table 6 show that all baselines have performance degradation when the scale enlarges. The $\mathcal{B}$-**Attention** obtains the best results in all parts; $\mathbf{A}_{self}$ works better than GCN; all output features are much better than the original ones. Among the baselines with attention mechanisms, the ones using GCN-based networks have better performances. The GCN with $\mathbf{A}_{band}$ has the highest mAP scores. Figure 5 shows the ROC curves of GCN, "GCN w/ $\mathbf{A}_{self}$" and "GCN w/ $\mathbf{A}_{band}$" across all parts of MS-Celeb, from which we can also see the performance degradation with larger testsets. The performance degradation of "GCN w/ $\mathbf{A}_{band}$" is much smaller than the other two baselines, which is shown more clearly from the differences of AUC in Table 7. In Table 7, differences on AUC of two adjacent parts  (e.g., part1 - part3, part3 - part5, part5 - part7, part7 - part9) and their summation for each method are shown. It can be observed that the summation of the differences of "GCN w/ $\mathbf{A}_{band}$" is 1.49, which is only 58.30% and 67.26% of the ones of GCN (2.56) and "GCN with $\mathbf{A}_{self}$" (2.22). This validates that the features from "GCN w/ $\mathbf{A}_{band}$" have a much smaller performance degradation than the other two baselines with larger testsets from the perspective of verification. Similar observations can also be seen in Table 6, where the performance degradation of the baselines with attention mechanisms is much smaller than GCN and original features.

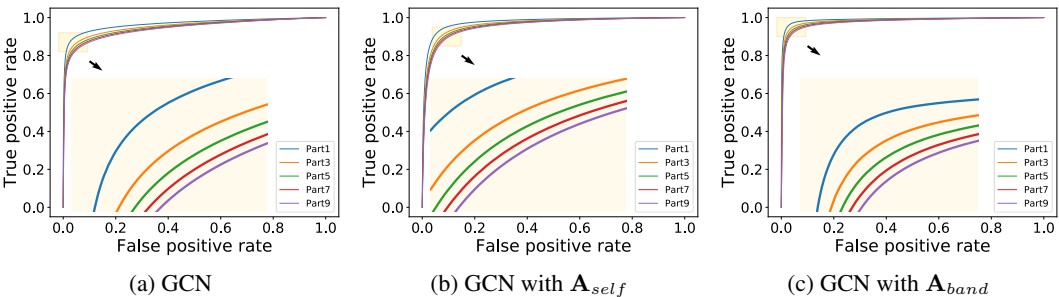

(a) GCN              (b) GCN with $\mathbf{A}_{self}$              (c) GCN with $\mathbf{A}_{band}$

Figure 5: ROC curves of GCN (a), GCN with $\mathbf{A}_{self}$ (b), and GCN with $\mathbf{A}_{band}$ (c) on MS-Celeb Part1-9.

Table 6: Feature quality (mAP) on different scales of MS-Celeb.

| Partition | Part1 | Part3 | Part5 | Part7 | Part9 |
|---|---|---|---|---|---|
| Original Feature | 80.07 | 75.28 | 73.21 | 71.73 | 70.63 |
| GCN | 90.15 | 86.27 | 84.34 | 82.95 | 81.90 |
| Transformer | 93.86 | 91.32 | 89.99 | 89.03 | 88.28 |
| GCN w/ $\mathbf{A}_{self}$ | 94.09 | 91.66 | 90.38 | 89.45 | 88.73 |
| Trans w/ $\mathbf{A}_{band}$ | 95.70 | 93.38 | 92.08 | 91.09 | 90.30 |
| GCN w/ $\mathbf{A}_{band}$ | **95.97** | **93.71** | **92.38** | **91.30** | **90.46** |

Table 7: Differences on AUC ($\delta_{AUC}$) of two adjacent parts of MS-Celeb for GCN, "GCN with $\mathbf{A}_{self}$" and "GCN with $\mathbf{A}_{band}$". The values in the table are scaled by a factor of 100 for better display.

| $\delta_{AUC}$ | GCN | GCN with $\mathbf{A}_{self}$ | GCN with $\mathbf{A}_{band}$ |
|---|---|---|---|
| part1 - part3 | 1.23 | 1.03 | 0.58 |
| part3 - part5 | 0.58 | 0.51 | 0.37 |
| part5 - part7 | 0.43 | 0.37 | 0.30 |
| part7 - part9 | 0.32 | 0.31 | 0.25 |
| summation | 2.56 | 2.22 | 1.49 |

## A.4    Ablation study on $\mathcal{B}$-Attention

Ablation experiments are conducted mainly on MS-Celeb to study the design of the topological architecture and attention fusion method of $\mathcal{B}$-**Attention**. Similar to Section 3.3, models are trained on Part0 and tested on Part1. All settings other than the factor being investigated are the same as that in Section 3.3 for "GCN with $\mathbf{A}_{band}$" with $k = 120$.

### A.4.1    topological architecture design

This study compares the topological architecture of the $\mathcal{B}$-**Attention** with three other designs: (1) one with only the $\mathbf{A}_{qart}$ of $\mathcal{B}$-**Attention**; (2) one using only the $\mathcal{Q}$-**Attention** part but with the $\mathbf{A}_X$ replaced by $\mathbf{A}_{self}$ in Equation 8, denoted as $\widetilde{\mathbf{A}}_{qart}$; (3) and one with $\mathcal{B}$-**Attention** but with the $\mathbf{A}_{qart}$ replaced by $\widetilde{\mathbf{A}}_{qart}$, denoted as $\widetilde{\mathbf{A}}_{band}$. Table 8 shows their mAP results, where the results of "GCN w/ $\mathbf{A}_{band}$" and one with only $\mathbf{A}_{self}$ are also listed for comparison. For the purpose of ablation study, the one with only $\mathbf{A}_{self}$ here has the same depth (i.e., three layers) with "GCN w/ $\mathbf{A}_{band}$", which is different from that in Table 2 (two layers). The comparisons are made on two conditions: with or without $\mathbf{W}_{qart}$ (i.e., $\mathbf{W}_{qart}^Q$ and $\mathbf{W}_{qart}^K$) in Equation 8. For the cases without $\mathbf{W}_{qart}$, $\mathbf{W}_{qart}^Q$ and $\mathbf{W}_{qart}^K$ are both identity matrices, i.e., $\mathbf{Q}_{qart} = \mathbf{K}_{qart} = \mathbf{A}_X$. Their ROC curves are shown in Figure 6. It can be seen that the $\mathcal{B}$-**Attention** design achieves the best results in both mAP and ROC; $\mathbf{A}_{qart}$ has better performance than $\mathbf{A}_{self}$; $\widetilde{\mathbf{A}}_{qart}$ and $\widetilde{\mathbf{A}}_{band}$ are worse than $\mathbf{A}_{qart}$ and $\mathbf{A}_{band}$ respectively. This shows that the combination of $\mathbf{A}_{self}$ and $\mathbf{A}_{qart}$ and the use of $\mathbf{A}_X$ (instead of $\mathbf{A}_{self}$) in $\mathbf{A}_{qart}$ both contribute to the performance improvement. In addition, the baselines with

$\mathbf{W}_{qart}$ have better performances (higher mAP scores and better ROC) than those without, indicating that $\mathbf{W}_{qart}^Q$ and $\mathbf{W}_{qart}^K$ are also critical for better graph structure learning.

Table 8: Feature quality (mAP) of different topological architectures of the attention mechanism.

| Arch./Cond. | w/ $\mathbf{W}_{qart}$ | w/o $\mathbf{W}_{qart}$ |
|---|---|---|
| $\mathbf{A}_{self}$ | 88.10 | 88.10 |
| $\widetilde{\mathbf{A}}_{qart}$ | 90.10 | 89.40 |
| $\mathbf{A}_{qart}$ | 92.26 | 88.92 |
| $\widetilde{\mathbf{A}}_{band}$ | 94.19 | 89.53 |
| $\mathbf{A}_{band}$ | **95.97** | **93.67** |

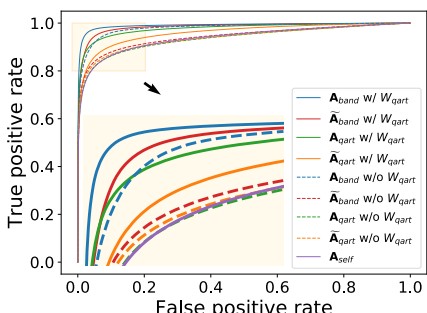

Figure 6: ROC curves of different topological architectures of the attention mechanism.

The ablation experiments are also **conducted on attention map fusion method in Section A.4.2 in Appendix**. The results in Table 9 and Figure 7 together will prove that the proposed form in Equation 11 has the best performance across all datasets.

### A.4.2 Attention map fusion method

The attention map fusion method is studied by comparing it to two other designs: direct element-wise addition without weighting ($\mathbf{A}_{qart} + \mathbf{A}_{self}$) and element-wise multiplication ($\mathbf{A}_{qart} \odot \mathbf{A}_{self}$) in Equation 11. The results are shown in Table 9 and Figure 7: The $\theta_{qart}\mathbf{A}_{qart} + \theta_{self}\mathbf{A}_{self}$ design works the best, and the performance of $\mathbf{A}_{qart} \odot \mathbf{A}_{self}$ is very close to $\theta_{qart}\mathbf{A}_{qart} + \theta_{self}\mathbf{A}_{self}$. However, results of MSMT17 and VeRi-776 in Table 9 shows our proposed method has the best performance across all datasets.

Table 9: Feature quality (mAP) of different attention map fusion methods, where $\odot$ denotes element-wise multiplication.

| Dataset | $\mathbf{A}_{qart} + \mathbf{A}_{self}$ | $\mathbf{A}_{qart} \odot \mathbf{A}_{self}$ | $\theta_{qart}\mathbf{A}_{qart} + \theta_{self}\mathbf{A}_{self}$ |
|---|---|---|---|
| MS-Celeb | 92.83 | 95.71 | **95.97** |
| MSMT17 | 84.95 | 85.59 | **86.04** |
| VeRi-776 | 85.97 | 85.53 | **87.59** |

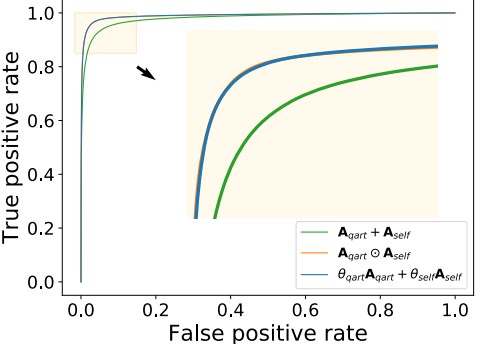

Figure 7: ROC curves of different fusion methods on MS-Celeb (part1).

Table 10: Clustering performance on MSMT17.

| Dataset | MSMT17 | |
|---|---|---|
| Metrics | $F_P$ | $F_B$ |
| K-Means | 60.32 | 69.28 |
| HAC | 63.95 | 73.75 |
| DBSCAN | 48.23 | 53.90 |
| Clusformer | - | - |
| STAR-FC | - | - |
| Ada-NETS | 73.51 | 77.40 |
| Original+G-cut | 50.60 | 66.56 |
| Ours+G-cut | **74.93** | **79.91** |
| Original+Infomap | 71.33 | 79.16 |
| Ours+Infomap | **77.54** | **81.75** |

### A.5 Clustering performance on MSMT17

The clustering tasks are also evaluated on MSMT17 and the results are in Table 10.

## A.6 Ensemble Performance on ReID tasks

In addition to directly comparing with the commonly used rerank methods (e.g., QE [13], LBR [39], KR [67] and ECN [47]) in Section 4.4, the "GCN with $\mathbf{A}_{band}$" can also be combined with them. The commonly used rerank methods can utilize the enhanced features from "GCN with $\mathbf{A}_{band}$" for better ranking. The results are shown in Table 11. Since mAP$^R$ can provide a more comprehensive evaluation of the feature quality than R1 for ReID, the mAP$^R$ is discussed more often. It can be observed that new SOTA performances are obtained on both datasets. The mAP$^\mathbf{R}$ on MSMT17 is as high as 79.67 by "$\mathbf{A}_{band}$+KR" and on VeRi-776 is as high as 85.92 by "$\mathbf{A}_{band}$+LBR". For most of the commonly used rerank methods, the mAP$^\mathbf{R}$ of reranking with $\mathbf{A}_{band}$ is higher than with original features and also higher than "GCN w/ $\mathbf{A}_{band}$". The only exception is "$\mathbf{A}_{band}$+ECN" which is higher than "GCN w/ $\mathbf{A}_{band}$" but only comparable with "original+ECN". The results show that there is a complementary relationship between "GCN with $\mathbf{A}_{band}$" and commonly used rerank methods. The "GCN with $\mathbf{A}_{band}$" can provide better features so that the commonly used rerank methods can be further improved, although the $R1$ results are only comparable.

Table 11: Ensemble ReID performance on MSMT17 and VeRi-776. "Original+X" means reranking with original features by methods of "X". "$\mathbf{A}_{band}$+X" means reranking with features from "GCN w/ $\mathbf{A}_{band}$" by methods of "X".

| Dataset | MSMT17 | | VeRi-776 | |
|---|---|---|---|---|
| Metrics | mAP$^R$ | R1 | mAP$^R$ | R1 |
| Original Feature | 62.76 | 82.36 | 79.00 | 96.60 |
| GCN w/ $\mathbf{A}_{band}$ | 78.28 | 86.64 | 84.74 | 96.78 |
| Original+QE | 71.10 | 84.37 | 83.09 | 95.47 |
| Original+LBR | 70.36 | 85.28 | 83.63 | **97.13** |
| Original+KR | 76.46 | 85.48 | 82.32 | 96.84 |
| Original+ECN | 78.79 | 86.16 | 84.25 | **97.13** |
| $\mathbf{A}_{band}$+QE | 79.37 | **86.91** | 85.88 | 96.90 |
| $\mathbf{A}_{band}$+LBR | 79.54 | 86.86 | **85.92** | 96.90 |
| $\mathbf{A}_{band}$+KR | **79.67** | 84.81 | 85.45 | 96.60 |
| $\mathbf{A}_{band}$+ECN | 78.70 | 84.05 | 85.61 | 96.90 |

## A.7 Declaration of the type of computing resources

All experiments are conducted on a single machine, with 54 E5-2682 v4 CPUs, 8 NVIDIA P100 cards and 400G memory.

# B Limitations

Our approach achieves promising results in graph structure learning, but it still has the following limitations:

- The $\mathcal{B}$-**Attention** requires more computation and number of parameters as shown in Section 2.3. However, the baseline methods can not achieve the same performance even adding their network depth. The layer number of network in each method is tuned in all experiments.

- The superiority of $\mathcal{B}$-**Attention** is not obvious enough on datasets with few samples in each category compared with self attention as revealed in Section 3.3.

- The performance of $\mathcal{B}$-**Attention** will also degrade when with large ENR as discussed in Section 3.3, indicating the current form still has improvement room on the robustness against noise to further investigate.

- The outputs $\mathcal{B}$-**Attention** will vary depending on the choice of nearest neighbours. However, the superiority of this method can be guaranteed as long as the conditions in Section 2.2 are met.

## C   Potential Negative Societal Impacts

The proposed graph structure learning method inherently is harmless just like many other AI technologies. This technology may have negative societal impacts if someone uses it for malicious applications such as surveillance, personal information collections. As authors, we advocate for proper technology usage.

## D   Pseudocode for self-attention and $\mathcal{Q}$-Attention

---

**Algorithm 1** PyTorch-style pseudocode for self-attention and $\mathcal{Q}$-**Attention**

```
# X: the input feature matrix, with shape (N, D)
def self_attention(X):
    Q_self = torch.matmul(X, W_self_Q)
    K_self = torch.matmul(X, W_self_K)
    self_attn = torch.matmul(Q_self, K_self.transpose(-2, -1))
    return self_attn

def Q_attention(X):
    A_X = torch.matmul(X, X.transpose(-2, -1))
    Q_qart = torch.matmul(A_X, W_qart_Q)
    K_qart = torch.matmul(A_X, W_qart_K)
    Q_attn = torch.matmul(Q_qart, K_qart.transpose(-2, -1))
    return Q_attn
```

---

## E   Detailed Proof of with generalized version of Chernoff Bounds

Below is a full analysis for the practical algorithm in Section 2.3 that does not assume a binary relationship between two nodes in $A_X$.

Let $X_i$ denote a random variable with no distribution assumptions such that $a \leq X_i \leq b$ for all $i$. Define $X = \sum_1^m X_i$, $\mu = \mathbb{E}(X)$. Then for all $0 < \epsilon < 1$, we have the general version of Chernoff inequality as follows.

($i$)  **Upper Tail:**  $\mathbb{P}(X \geq (1+\epsilon)\mu) \leq e^{-\frac{\epsilon^2 \mu^2}{m(b-a)^2}}$ ;

($ii$)  **Lower Tail:**  $\mathbb{P}(X \leq (1-\epsilon)\mu) \leq e^{-\frac{\epsilon^2 \mu^2}{m(b-a)^2}}$ .

The above show that with probability at least $1 - \delta$, $(1-\epsilon)\mu \leq X \leq (1+\epsilon)\mu$ where $\delta = e^{-\frac{\epsilon^2 \mu^2}{m(b-a)^2}}$ or equivalently $\epsilon = \sqrt{\frac{m(b-a)^2 log(1/\delta)}{\mu^2}}$.

Instead of assuming $p$ and $q$ for making connections in the case of $C(v_i) = C(v_j)$ and $C(v_i) \neq C(v_j)$, we assume that the similarity score $s_{i,j}$ is sampled from a bounded distribution (bounded between -1 and +1) with mean of $s_+$ when $C(v_i) = C(v_j)$, and $s_{i,j}$ is sampled from a bounded distribution (bound between -1 and +1) with mean of $s_-$ when $C(v_i) = C(v_j)$. We of course assume $s_+ > s_-$. In expectation, the number of edges connected under $S_{i,j}^k$ on $\mathcal{V}_{i,j}$ is

$$
\begin{cases}
\mu^- = s_+ s_- m, & C(v_i) \neq C(v_j), \\
\mu^+ = (\alpha s_+^2 + (1-\alpha)s_-^2)m, & C(v_i) = C(v_j),
\end{cases}
\tag{12}
$$

where $C(v)$ is the category of node $v$.

It should be noted that $(b-a)^2 = 4$ under our assumption. Therefore, when $v_i, v_j$ are sampled from different categories, we have, with probability at least $1 - \delta$,

$$\sum_{k \in V_{i,j}} S_{i,j}^k \leq (1 + \epsilon^-)\mu^-$$

$$\sum_{k \in V_{i,j}} S_{i,j}^k \leq \left(1 + \sqrt{\frac{4mlog(1/\delta)}{(\mu^-)^2}}\right)\mu^- \tag{13}$$

$$S_{i,j} = \frac{1}{m}\sum_{k \in V_{i,j}} S_{i,j}^k \leq \left(1 + \sqrt{\frac{4nlog(1/\delta)}{(s_+ s_- m)^2}}\right) s_+ s_- \ .$$

When $v_i, v_j$ are sampled from the same category, we have, with probability at least $1 - \delta$,

$$\sum_{k \in V_{i,j}} S_{i,j}^k \geq (1 - \epsilon^+)\mu^+$$

$$\sum_{k \in V_{i,j}} S_{i,j}^k \geq \left(1 - \sqrt{\frac{4mlog(1/\delta)}{\mu^+}}\right)\mu^+ \tag{14}$$

$$S_{i,j} = \frac{1}{m}\sum_{k \in V_{i,j}} S_{i,j}^k \geq \left(1 - \sqrt{\frac{4nlog(1/\delta)}{(\alpha s_+^2 + (1-\alpha)s_-^2)^2 m^2}}\right)(\alpha s_+^2 + (1-\alpha)s_-^2) \ .$$

The two distributions can be well separated if **the above two bounds do not overlap**. That is, we require that

$$(1 + \epsilon^-)\mu^- < (1 - \epsilon^+)\mu^+ \ . \tag{15}$$

On the other hand, it is obvious $s_+ s_- < \alpha s_+^2 + (1-\alpha)s_-^2$, so we know

$$\epsilon^- = \sqrt{\frac{4mlog(1/\delta)}{(s_+ s_- m)^2}} > \epsilon^+ = \sqrt{\frac{4mlog(1/\delta)}{(\alpha s_+^2 + (1-\alpha)s_-^2)^2 m^2}} \ . \tag{16}$$

Set $\epsilon = \epsilon^-$ to have a larger step and a stricter condition, we have

$$(1 + \epsilon)\mu^- < (1 - \epsilon)\mu^+$$
$$\epsilon < \frac{u^+ - u^-}{u^+ + u^-} \tag{17}$$

$$\sqrt{\frac{4m\log(1/\delta)}{(s_+ s_- m)^2}} < \frac{(s_+ - s_-)^2 + (2\alpha - 1)(s_+^2 - s_-^2)}{(s_+ + s_-)^2 + (2\alpha - 1)(s_+^2 - s_-^2)} . \tag{18}$$

Because $\alpha > \frac{1}{2}$ and $s_+ > s_-$,

$$(s_+ - s_-)^2 + (2\alpha - 1)(s_+^2 - s_-^2) > 0,$$
$$(s_+ + s_-)^2 + (2\alpha - 1)(s_+^2 - s_-^2) > 0. \tag{19}$$

So we have

$$m > \frac{4((s_+ + s_-)^2 + (2\alpha - 1)(s_+^2 - s_-^2))^2}{s_+^2 s_-^2 ((s_+ - s_-)^2 + (2\alpha - 1)(s_+^2 - s_-^2))^2})\log(\frac{1}{\delta}). \tag{20}$$

We complete the proof by substituting $\delta$ with

$$\delta \triangleq \frac{1}{\gamma}\min((1-\alpha)s_-, \alpha(1-s_+)). \tag{21}$$

where $\gamma > 1$ is a constant.

Finally, we prove that

$$m > \frac{4((s_+ + s_-)^2 + (2\alpha - 1)(s_+^2 - s_-^2))^2}{s_+^2 s_-^2 ((s_+ - s_-)^2 + (2\alpha - 1)(s_+^2 - s_-^2))^2} \log \left( \frac{\gamma}{\min((1-\alpha)s_-, \alpha(1-s_+))} \right) . \quad (22)$$

We can guarantee that when the above condition is met, the proposed similarity measure between two nodes from the same category is strictly larger than that for two nodes from different categories. It is able to reduce the number of noisy edges and missing edges of the single test method by $1/\gamma$ in expectation.

## F  Case Studies on self attention, $\mathcal{Q}$-Attention and $\mathcal{B}$-Attention

To better understand the effect of multiple tests (i.e. the fourth order of statistics), we illustrate the effect of self-attention, $\mathcal{Q}$-**Attention** and $\mathcal{B}$-**Attention** with the help of the tool BertViz[5].

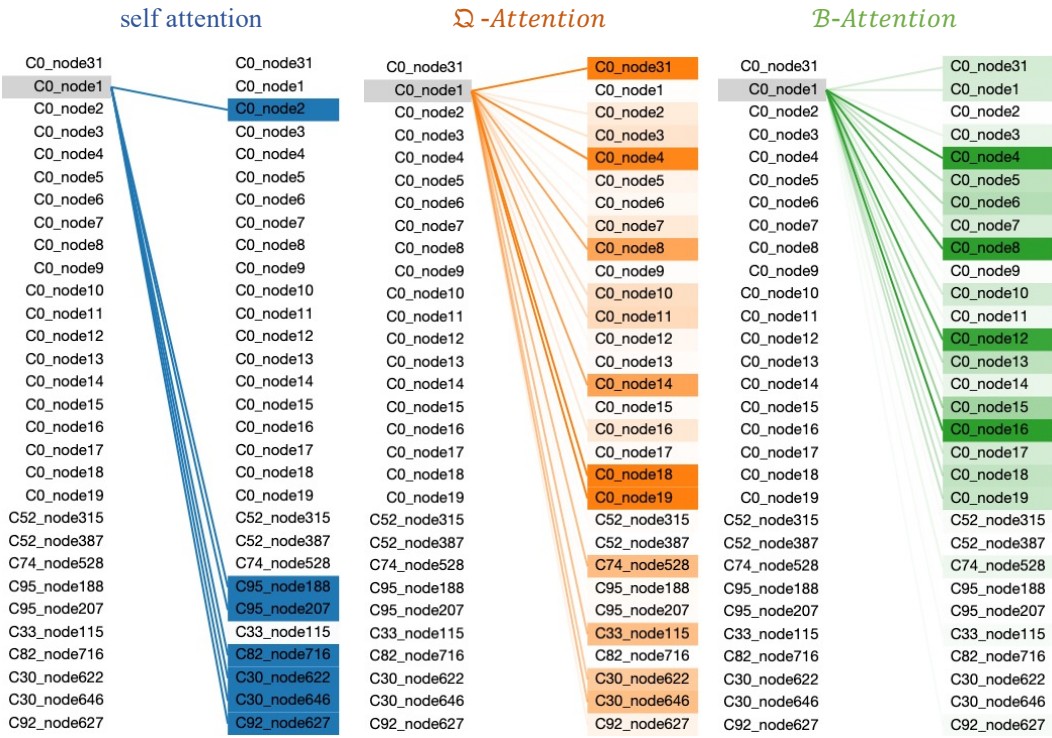

Figure 8: Case studies on self attention, $\mathcal{Q}$-**Attention** and $\mathcal{B}$-**Attention**. Each entry contains the category and node index, e.g. "C0_node1" means this node belongs to category 0 and the node index is 1. The shade of the colour represents the weight of attention. The darker the colour, the greater the weight. The data of this case is sampled from the last layer of $\mathcal{B}$-**Attention** on the MS-Celeb part1.

As shown in the left panel of Figure 8, self-attention introduces a number of noisy connections between nodes belonging to different categories. In contrast, according to the middle panel of Figure 8, $\mathcal{Q}$-**Attention** is able to introduce many more connections between nodes from the same category, and at the same time significantly reduces the weights for those noisy connections. In the last panel, the combination self-attention with the fourth order statistics, The $\mathcal{B}$-**Attention** further removes those noisy connections while keeps most of the clean connections between nodes.

---

[5]https://github.com/jessevig/bertviz