# OpenReview forum: "Robust Graph Structure Learning via Multiple Statistical Tests"
_NeurIPS.cc/2022/Conference — NeurIPS 2022 Accept_

### Official Review · Reviewer_1q9s · 2022-07-09

**Rating:** 4
**Confidence:** 4
**Soundness:** 3 good
**Presentation:** 3 good
**Contribution:** 2 fair

**Summary:**

The core target of this paper aims to use GCNs with the common neighbor sets to alleviate the impact of noises in expectation, through modeling the vision tasks as the *node classification*.

At first, the authors theoretically show that using the common node set $\mathcal V_{ij}$ to compute the similarity of $v_i$ and $v_j$,  would help to reduce the variance of the estimation of pairwise nodes. Then, similar to the popular self-attention (which is widely used in Transformer), two graph matrices, $A_{band}$ and $A_{self}$, are designed. Some experiments of ReID and clustering are conducted.

**Questions:**

1. What's the connection between Eq. 10 and $k$NN? How does the model obtain the sparse graph (as the authors claim in Line 179)?

2. Is the model expensive, from both time and space perspectives? It is indeed an intrinsic problem of graph models, but the authors should also consider the negative impact.

3. Could the authors please clarify the connection between Section 2.2 and Section 2.3 more formally?

**Ethics Review Area:**

["I don’t know"]

**Limitations:**

It has been discussed in the appendix.

**Strengths And Weaknesses:**

### Pros:

1. The paper is well-written and easy to follow.
2. The motivation is technically sound. Using more nodes/points to reduce the variance is a reliable scheme.

### Cons:

1. As the vision task is viewed as the *node classification*, the size of graph would increase with the size of the dataset. It indicates that the inefficiency problem is an inescapable (and severe) problem. However, there is no discussion in Eq. 8-11. It seems that the model computes a dense similarity matrix, while the authors also claim that the graph is constructed by $k$NN. What's the connection between Eq. 10 and $k$NN? Does it compute $A_{band}$ firstly and then find a $k$NN-approximation matrix that only retains the $k$  largest entries?  Besides, $A_X=XX^T$ is already time-consuming.  It would be better if the authors provide the pseudo-code of the whole procedure in the appendix.

2. Although the motivation to reduce the impact of noises is convincing, the design in Section 2.3 seems to imitate the self-attention mechanism. In other words, it seems a little irrelevant to Section 2.2 since Eq. 8-11 seems not to be strongly driven by the theoretical results. It looks like Theorem 1 is the explanation of Eq. 8-11, rather than the motivation.

3. The conclusion of Theorem 1 should be more formal. In other words, it should be described mathematically. What's more, Line 79-81 should appear as a lemma.

4. The GCN formulation in Eq. 1 is not standard. The authors should clarify it.

5. I suggest using $\mathcal{Q}$-Attention and $\mathcal B$-Attention.

Overall, my main concerns come from two aspects, the inefficiency problem and the weak connection between Sections 2.2 and 2.3. I would like to update my score after reading other reviews and the response.

---

> ### Author Response · Authors · 2022-08-02
> **Response to Reviewer 1q9s**
>
> **Q1. [The construction of graphs and high computational complexity for large graphs.]**
>
> Two tricks are used in our study to make the computation scalable to large datasets.
> First, similar to existing works (Wang et al., 2019; Yang et al., 2020; Guo et al., 2020; Nguyen et al., 2021; Shen
> et al., 2021; Wang et al., 2022; Zhong et al., 2017; Sarfraz et al., 2018), $k$NN is used to construct the initial sparse graph, which is $A_X$ in Eq. 8. Second, even with $k$NN, the resulted graph size can be very large for large datasets. We thus adopted the subgraph sampling approach from deep graph library framework (DGL) (Wang M. et al., 2019), which allows us to work on subgraphs instead of the entire graph, making our approach scalable to datasets with millions of nodes. More specifically, we set the number of nodes in each sampled subgraph to be $256$, and as a result, our matrix $A_X$ is of size $256\times 256$, a small matrix to handle in computation. Pseudo codes for self-attention and $\mathcal{Q}\mbox{\bf ttention}$ are provided in **Algorithm** 1 of **Section D in Appendix**.
> In addition, we tuned the number of layers for all the models for their best performance in experiments.
> As shown in Line 236-237, $A_{band}$ uses one layer while Transformer and GCN use three layers in VeRi-776 and MSMT17. However, $A_{band}$ has the best results as in Table 2.
> We will make it clear in the revised version.
>
>
> **Q2. [The connection between Theoretical Analysis in Section 2.2 and Practical Algorithm in Section 2.3, and Importance of Multiple Tests vs. Self-Attention.]**
>
> 1. In Section 2.2, for the sake of simplicity, we assumes a binary relationship between any two nodes in matrix $A_X$, where in Section 2.3 $A_X$ is a matrix of real numbers and is computed based on dot products between feature vectors and the $k$NN graph. The reviewer is concerned if the theoretical result can be applicable to the practical algorithm given in Section 2.3. Here, we provide affirmative answer to this question. Attached in **Section E of Appendix** you can find more detailed analysis that extends the results of Theorem 1 to the practical algorithm. The key is to leverage the generalized version of Chernoff inequality that allows for multiple tests with real numbers. The conclusion is similar to that of Theorem 1, i.e. with large enough nearest neighbors used for multiple tests, in expectation, we can significantly reduce the number of noisy edges in the graph.
>
> 2. Although we introduce self-attention into the practical algorithm, as indicated in our empirical studies (i.e. Table 2 and 4), the self-attention mechanism only provides marginal improvements compared to the baseline methods. It is the introduction of $A_{qart}$, the fourth order of statistics, that leads to significant more improvements in our empirical studies. We further provide case studies in the response to Reviewer 1 (Section 2.3) in **Section F of Appendix** that highlight the impact of multiple tests versus self-attention.
>
> **Q3. [Suggestions for presentation.]**
>
> We thank the reviewer for suggestions on presentation, particularly making presentation in Theorem 1 more rigorously, and better naming for variables and algorithms. We will revise accordingly in the final version.

---

> > ### Comment · Reviewer_1q9s · 2022-08-07
> > **Response to authors**
> >
> > Thanks for your response. The provided response has partially addressed my concerns.
> >
> > I strongly suggest that the authors should clearly point out that the sub-graph sampling strategy is used, especially in Section 2. The efficiency problem is pretty crucial for GNN and it is not a dispensable "training setting". How to efficiently and stably train GNN on large graphs is still a problem to be solved.

---

> > > ### Author Response · Authors · 2022-08-08
> > > **Response to Reviewer 1q9s**
> > >
> > > Thanks for the suggestion! The manuscript has been updated by adding a short paragraph (in red color) in Section 2 to explicitly describe the sub-graph sampling strategy for training and inference.
> > >
> > > The sub-graph sampling strategy we adopted is similar to that in DGL (Wang et al., 2019) and Ada-NETS (Wang et al., 2022):
> > > In particular, $k_{seed}$ nodes are first selected as seeds from a dataset in each sampling step. Then the seed nodes together with their $k$ nearest neighbours ($k$NN) form the nodes of a sub-graph.
> > > As a result, Equations 8 and 9 are applied to the sub-graph during training and inference.
> > > For a dataset of size $N_{d}$, it takes about $N_{d}/k_{seed}$ iterations to process the whole dataset on average.
> > > More implementation details and settings will be included in the code to be open-sourced.
> > >
> > > We agree that how to efficiently and stably train GNNs on large graphs is still an open question. We are interested and also planning to do more studies on this in our future work.

---

### Official Review · Reviewer_Xw91 · 2022-07-09

**Rating:** 8
**Confidence:** 4
**Soundness:** 4 excellent
**Presentation:** 3 good
**Contribution:** 4 excellent

**Summary:**

The paper attempts to build a robust graph structure learning approach over images from the viewpoint of multiple statistical tests. The method is novel and proven theoretically according to the Chernoff Bounds. The corresponding elegant matrix form of that method, named Bttention, is designed for efficiency. The experimental results are solid, and the analysis and conclusions are very sufficient. The story sounds interesting and meaningful. It is a fundamental research to the NeurIPS community.

**Questions:**

(1) How to choose the optimal k value for new datasets?
(2) The reviewer is concerned about whether the code will be made public.


**Limitations:**

The authors have discussed the limitations of this paper in detail in the appendix, which is already comprehensive.

**Strengths And Weaknesses:**

Strengths:
(1) The effectiveness of multiple tests for graph structure learning is verified both theoretically and empirically on multiple clustering and ReID benchmark datasets. The originality is sufficient.
(2) The Bttention is well-designed in the elegant matrix form which is easy to implement. A high quality contribution on technology.
(3) The logical flow is clear. It first indicates the unreliability of pairwise similarity, then proposes their Sim-M and Bttention in Section 2. Finally, it validates their robustness to noise and shows SOTA performances on downstream tasks in Section 3.
(4) The experiments are very sufficient and well organized. It significantly outperforms previous SOAT methods in clustering and ReID tasks. These results can support the conclusions.

Weaknesses:
(1) This method cannot be adapted to various datasets. As proven in Section 2.2 and empirically shown in Section 3.3, it is only advantageous when the number of samples in each category is large enough in the dataset.
(2) The performance of Bttention will degrade when with too large ENR as discussed in Section 3.3.

---

> ### Author Response · Authors · 2022-08-02
> **Response to Reviewer Xw91**
>
> **Q1. [The optimal choice of $k$ value for new datasets.]**
>
> We generally determine the optimal $k$ by tuning the parameters in the validation set.
> As shown in Figure 3, the proposed method is robust to the average Edge Noise Rate (ENR) which is controlled by $k$.
> So our method has a relatively large optimal range in the value of $k$.
> It should be noted that if the dataset has a small average sample amount for each individual, the optimal $k$ value needs to be carefully tuned on the validation set.
>
>
> **Q2. [The open source code of this paper.]**
>
> We are glad to contribute to the NeurIPS community and will release the code and datasets after the paper is accepted.
> Please keep up with our work.
>
> **Q3. [The two weaknesses mentioned by the reviewers.]**
>
> Regarding the two weaknesses mentioned by the reviewer, we have already stated them in the Limitations Section in Appendix. We will try to address these weaknesses in our future work.

---

### Official Review · Reviewer_dFZi · 2022-07-13

**Rating:** 7
**Confidence:** 4
**Soundness:** 3 good
**Presentation:** 3 good
**Contribution:** 3 good

**Summary:**

This paper proposes a graph structure learning for images that have no graph structure naturally. Instead of simply computing the pairwise similarity of images to form the edge, This paper proposes to leverage a single statistical test to form the edges between images, namely Sim-S.  In addition, this paper proposed to leverage the multiple statistical tests for robust graph structure learning, namely Sim-M.  Bttention is also proposed to craft the proposed method into an elegant and efficient matrix form for efficacy in real-world tasks. The experiments demonstrate the effectiveness of the proposed method, Sim-S, Sim-M, and Bttention.

**Questions:**

1. I would like to see the potential solution to reduce the time of the fourth-order statistics in the **Weaknesses 1**
2. As footnote 1 said, “all the datasets used in this paper are feature vectors” (Page 5).  Thus I wonder how the proposed method performs while it uses the original features. Since the learned features would be meaningful and less noisy than the original features.
3.  I would like to see some case studies to show the results of the proposed method, for example, a small graph learned in the real-world graph.
4. I would suggest that there should be more introduction to Statistical Tests in Section 2.1.


**Ethics Review Area:**

["I don’t know"]

**Strengths And Weaknesses:**

**Strengths**

1. This paper is well-motivated. Since the simple similarity between images can not often capture the relation between the image pairs, thus a more accurate way to compute the similarity between image pairs is needed.
2. The proposed method is technically sound.  The effectiveness of the single test and multiple tests is well supported by the theoretical analysis in  Proposition 1 and Theorem 1.  The proposed method is also practical since it provides an elegant and efficient matrix form.
3. The experimental results are convincing to me. Sections 3.3 and 3.4 show the effectiveness of the proposed methods, also the robustness of Sim-M to noise.  Table 3 shows the superiority of the proposed methods for clustering tasks. This paper also demonstrates the superiority of the proposed method on downstream tasks.
4. The paper is well organized and easy to follow. The proposed method is easy to implement and reproduce.

**Weaknesses**

1. Although the proposal is simple and practical, fourth-order statistics of features are used in Bttention. Such an operation would be time-consuming.
2. As “Better performance can not be achieved by enlarging k values for MSMT17, as its average sample amount 251 for each individual is only 31” (Line 251) said,  the performance of Bttention will go bad in some cases, and this paper didn’t provide the insightful analysis for this phenomenon.
3. The Preliminary part lacks the introduction of  Statistical Tests. And the Chernoff Bounds part in the section is not clear. The usage and advantages of this term should be discussed.

---

> ### Author Response · Authors · 2022-08-02
> **Response to Reviewer dFZi**
>
> **Q1. [The Computational Efficiency of Fourth-order Statistics.]**
>
> We have discussed computational efficiency for $A_{self}$ and $A_{qart}$ in the response to Reviewer 1q9s (the answer to **Q1**).
> Please refer to it for more details.
> The essential ideas are to construct sparse graph using $k$NN and reduce the size of $A_X$ by sampling sub-graphs of small size ($256\times 256$ in our case). Pseudo codes are provided in Algorithm 1 of **Section D in Appendix**.
>
>
> **Q2. [Experimental Results for Using Original Features.]**
>
> Image features used in our studies are extracted by CNN models and are the original features coming together with the datasets. To ensure a fair comparison with other baselines, we did not tune image features in our study.
>
>
> **Q3. [Illustrations of Power of Multiple Tests from Case Studies.]**
>
> To better understand the effect of multiple tests (i.e. the fourth order of statistics), we illustrate the effect of self-attention, $\mathcal{Q}\textbf{ttention}$ and $\mathcal{B}\textbf{ttention}$ with the help of the tool BertViz in **Figure 8 of Appendix**.
>
> As shown in the left panel of Figure 8 in Appendix, self-attention introduces a number of noisy connections between nodes belonging to different categories. In contrast, according to the middle panel of Figure 8, $\mathcal{Q}\textbf{ttention}$ is able to introduce many more connections between nodes from the same category, and at the same time significantly reduces the weights for those noisy connections. In the last panel, the combination of self-attention with the fourth order statistics, the $\mathcal{B}\textbf{ttention}$ further removes those noisy connections while keeps most of the clean connections between nodes of the same category.
>
> **Q4. [Further Analysis to Confirm the Statement on Line 251.]**
>
> Since the accuracy of multiple tests improves with increasing number of tests, which is $k$ in the $k$NN graph, it is tentative to increase $k$. However, on the other hand, our analysis assumes that for any node $v$, there are more nodes in the $k$ nearest neighbors sharing the same category as $v$ than those sharing different categories (i.e. $\alpha > 1/2$). By significantly enlarging $k$, we will end up with $\alpha < 1/2$ in $k$ nearest neighbors, which fails the assumption of our algorithm. The optimal $k$ is determined by a validation set.
>
>
> We thank the reviewer for suggestions on presentation, and we will revise accordingly in the final version.

---

### Author Response · Authors · 2022-08-02
**General Response**

We thank the reviewers for their insightful comments and suggestions.
Please find attached below our responses to the questions and concerns raised by different reviewers.

---

### Meta-Review · Area_Chair_usPA · 2022-08-31

**Recommendation:** Accept
**Confidence:** Certain

**Metareview:**

All reviews are quite positive and raised concerns have been addressed.

Accept.

**Award:**

No

---

### Decision · Program_Chairs · 2022-09-14

Accept